# Stimulus-Responsive Ultrathin Films for Bioapplications: A Concise Review

**DOI:** 10.3390/molecules28031020

**Published:** 2023-01-19

**Authors:** Maria Benelmekki, Jeong-Hwan Kim

**Affiliations:** 1Nanomaterials Lab, College of Engineering, Swansea University, Bay Campus, Fabian Way, Swansea SA1 8EN, UK; 2Cardiovascular Research Institute, Graduate School of Medicine, Yokohama City University, Yokohama 236-0004, Japan

**Keywords:** 2D materials, hybrid nanosheets, stimulus-responsive, nanomedicine, nanotechnology, surface chemistry, interfaces

## Abstract

The term “nanosheets” has been coined recently to describe supported and free-standing “ultrathin film” materials, with thicknesses ranging from a single atomic layer to a few tens of nanometers. Owing to their physicochemical properties and their large surface area with abundant accessible active sites, nanosheets (NSHs) of inorganic materials such as Au, amorphous carbon, graphene, and boron nitride (BN) are considered ideal building blocks or scaffolds for a wide range of applications encompassing electronic and optical devices, membranes, drug delivery systems, and multimodal contrast agents, among others. A wide variety of synthetic methods are employed for the manufacturing of these NSHs, and they can be categorized into (1) top-down approaches involving exfoliation of layered materials, or (2) bottom-up approaches where crystal growth of nanocomposites takes place in a liquid or gas phase. Of note, polymer template liquid exfoliation (PTLE) methods are the most suitable as they lead to the fabrication of high-performance and stable hybrid NSHs and NSH composites with the appropriate quality, solubility, and properties. Moreover, PTLE methods allow for the production of stimulus-responsive NSHs, whose response is commonly driven by a favorable growth in the appropriate polymer chains onto one side of the NSHs, resulting in the ability of the NSHs to roll up to form nanoscrolls (NSCs), i.e., open tubular structures with tunable interlayer gaps between their walls. On the other hand, this review gives insight into the potential of the stimulus-responsive nanostructures for biosensing and controlled drug release systems, illustrating the last advances in the PTLE methods of synthesis of these nanostructures and their applications.

## 1. Introduction

### 1.1. Significance of Nanosheets

The term “nanosheets” was recently coined to describe supported and free-standing “ultrathin film” materials, with thicknesses ranging from a single atomic layer to a few tens of nanometers. However, their length and width can be measured in microns, depending on the method of synthesis. Within nanomaterial classifications, nanosheets (hereafter NSHs) are classified as two-dimensional materials (2D materials). Hence, the terms “2D materials”, “ultra-thin films”, and “nanosheets” are widely used as synonyms to describe materials that have one dimension in the nanoscale and are extended in the other two dimensions [1,2,3]. The physicochemical properties of these nanostructures and their large surface area with abundant accessible active sites make them attractive components for a wide range of applications encompassing electronic and optical devices, membranes, drug delivery systems, and multimodal contrast agents, among others [3,4,5,6,7,8]. NSHs can be made of organic (usually polymers and surfactants) or inorganic (e.g., Au, boron nitride (BN), graphene, and transition-metal dichalcogenides (TMDs) (MoS_2_, TaS_2_, WS_2_…)) materials [9,10,11,12,13,14,15,16,17,18,19]. When NSHs are composed of a combination of both organic and inorganic materials, they are commonly known as “hybrid NSHs”. Within the category of “hybrid NSHs”, another class of NSHs consisting of layered nanostructures is known as “Janus NSHs”. The constituting layers can be made of organic or inorganic materials, or a combination of both (more details are provided in Section 3).

These nanostructures present huge potential as chemical and biological sensing materials due to their physical and chemical properties, ease of functionalization, and flexibility. Moreover, when combined with appropriate polymers [15,16,17,18,19], NSHs acquire the capacity to respond to external stimulations such as temperature, light, or pH changes, thus undergoing a reversible shape transformation from 2D to 1D (one dimensional) materials, to form “nanoscrolls” (NSCs). NSCs present open tubular structures with tunable interlayer gaps between their walls. In addition, the tuning of the NSC dimensions enables them to trap specific molecules, ions, drugs, and biomolecules; thus, they can be used as biochemical sensors, drug delivery vehicles, and molecular imaging platforms [3,4,5,6,7,8,9,10,11,12,13].

The present review aims to give insight into the design and synthesis of NSHs with emphasis on inorganic and hybrid inorganic–organic NSHs. Different designs are presented including (i) inorganic NSHs (carbon, graphene, BN, and Au), and (ii) combinations of organic–inorganic materials, considering different structures such as composite, layered, and Janus structures. Polymers with different chemical properties are introduced to create stimulus-responsive NSHs. By adding the correct polymer, NSHs with a polymer sensitive to the acidity or basicity of the surrounding medium, such as polyethyleneimine (PEI), will result in pH-responsive NSHs, which could have great potential for targeting different parts of the human digestive tract, which changes pH between the acidic stomach and basic intestines, while a heat-sensitive polymer (e.g., poly(ε-caprolactone)-b-polyethylene glycol (PEG-b-PCL)) could burn surrounding tumors to destroy them, functioning as a kind of super-specific chemotherapy [5].

### 1.2. Insight into Nanomaterials and Nanostructures Applied to Medicine

Currently, nanomaterials and nanostructures attract great interest for a wide range of applications linked to public health and safety. For example, smart and active nanomaterials can be easily applied to goods and objects used in our daily routines, providing alternatives to environment preservation and remediation, and bringing notable improvements in public health when used for antibacterial, self-cleaning, and self-healing ends [15,16,20]. In the specific case of nanomaterials applied to medicine (nanomedicine), this multidisciplinary field has captured the interest of researchers and engineers from different disciplines, with aims to provide solutions for early diagnosis and targeted therapy, toward personalized medicine (Figure 1). Over the past decade, concepts and tools derived from nanotechnology, nanomaterials, and biotechnology have been applied to overcome the problems of conventional techniques for advanced diagnosis and therapy. Multidisciplinary research, bringing together physicists, chemists, biologists, and engineers, as illustrated in Figure 1, aims to improve sensing and imaging techniques for an early detection of pathological changes at the molecular level by means of clear and conclusive imaging methods and minimally invasive treatment of the patient. Therefore, putting together multidisciplinary skills is likely to be the most effective shortcut to build the required knowledge and “know-how” toward “accessible personalized medicine”. In particular, advances in nanomaterial technology, merging nanomaterials with different properties, have created new paradigms for multifunctional nanostructures within a single platform [14,15,16,17,18,19,20].

The expectations of the diagnostic, therapeutic, and regenerative possibilities of nanomedicine are projected in different paths such as inexpensive rapid tests for viral infection and the first signs of diseases long before symptoms manifest themselves, genetic predisposition, and medicines and vaccines without side-effects for treatments of cancer, cardiovascular diseases, and neurological diseases [20,21,22].

For example, novel nanostructures that combine photonic, plasmonic, and magnetic properties can lead to highly sensitive and cost-effective biosensors, allowing important improvements in patient care while reducing costs, contributing to the efficiency of the hospital logistics, and enhancing safety by allowing the early detection of specific biological markers (biomarkers) at a single-molecule level [23,24,25,26]. In addition, these multifunctional nanomaterials enable multimodal contrast agents that combine different properties such as magnetic and optical intrinsic responses, enhancing the patient’s safety by limiting the number of contrast agent administrations required for imaging purposes [26,27,28].

### 1.3. Application of Nanosheets (NSHs) in Nanomedicine

Generally speaking, the design of complex hybrid nanostructures that combine organic and inorganic materials with advanced functionalities, and the study of their fundamental properties have a major role in the development of a new generation of nanostructured materials. In the specific case of 2D materials, the possibility of tailoring the dimension, composition, and structure of hybrid NSHs represents a major milestone in the control of their physicochemical properties. These properties, combined with the ability to produce high quality NSHs, lead to their potential applicability in advanced nanomedicine [14,20]. The combination of different functionalities in a one-phase 2D nanostructured material is attractive for accurate and preventive diagnostic and prognostic tools. Engineering and assembling such nanostructures and integrating them into a single scaffold with controllable geometry, interface, and properties would lead to improved performance of diagnostic devices [23,24,25,29,30]. Other relevant applications are related to multifunctional contrast agents for advanced multimodal bioimaging [26,27,28,31,32].

Within the biological sensing field, nanotechnology has an important role in the development of more effective and multifunctional biosensors, leading to a better life quality and personalized knowledge of the patient [33,34,35,36,37,38,39,40,41,42,43]. Nanomaterials are playing an important role in the development of biosensors since they clearly increase the analytical performance. The improvements are mainly related to the increased surface area, allowing an enhanced accessibility for the analyte (compound to be detected) to the receptor unit (sensing element). Nanomaterials can also add value to biosensor devices due to their intrinsic physical or chemical properties and can even act as transducers for the signal capture. Among the vast number of examples where nanomaterials demonstrate their superiority to bulk materials, the combination of different nano-objects with different characteristics can create phenomena which contribute to new or improved signal capture setups [14,30,44,45].

Although biosensors have been actively studied to address current on-site detection or point-of-care demands in biomedical applications, their investigation in practical applications remains challenging. Currently, most of the reported results are performed in optimal laboratory settings as a proof-of-concept, instead of using complex environment (e.g., whole blood, urine, food, and cells), where the mixture of biomolecules and ions often induces false signals and reduces the sensitivity [40]. Therefore, more emphasis is placed on assessing feasibility and performance in complex environments. An example of biomolecule detection enhancement, using a combination of NSHs and plasmonic nanoparticles, is discussed in Section 3.2.

## 2. Methods of Synthesis of NSHs

The strategies employed for the manufacturing of NSHs can be categorized into top-down and bottom-up approaches. Generally speaking, both approaches are used in nanofabrication processes. However, in the case of top-down methods, the initial material is a bulk-material, and the processes are analogous to forming a “statue”, starting from a stone. The top-down approach involves exfoliation of layered materials using techniques such as ultrasonic irradiation and ball-milling [46,47]. Bottom-up methods instead consist of building up materials and structures from groups of atoms. Most bottom-up approaches allow crystal growth of nanocomposites in a liquid or gas phases [48,49].

### 2.1. Top-Down Synthesis

Top-down methods for the production of NSH materials can be divided into physical (also known as mechanical) and chemical exfoliation processes of their bulk counterpart. Most of these methods present considerable limitations such as harsh reaction conditions, high energy consumption, impurity contamination, and poor solubility, which hinder an efficient production of high-quality NSHs. Herein, we discuss the methods of synthesis of two of the most attracting layered materials for bioapplications: hexagonal boron nitride (hBN) and graphene [50,51]. Both materials have been in the research spotlight since the discoveries of their interesting properties, such as high mechanical strength, transparency, good thermal conductivity, and excellent chemical and thermal stabilities. Both materials have attracted significant interests in biological applications such as tumor labeling, sensing, and targeting [52]. However, carbon-based nanomaterials show higher toxicity in both in vivo and in vitro investigations, while BN nanostructures show a better biocompatibility and lower cytotoxicity in comparison to carbon nanostructures [52].

Regarding graphene, mechanical cleavage by Scotch tape was the first method to prepare free-standing graphene NSHs; thus, it was extended to other materials such as hBN and MoS_2_ [53,54,55]. During mechanical cleavage, the pulling force could easily break the weak Van der Waals interactions between graphene layers and leave the strongly sp^2^-bonded in-plane structure intact. The peeled NSHs present fewer defects, compared to those produced by chemical methods; however, the yield of production by this method is very low, hindering use for large studies and practical applications. Ball milling techniques were evaluated for production of graphene NSHs with a good quality and a fair yield. However, most ball milling treatments result in NSHs with disordered crystalline structure, not to mention the contamination of the final product [56,57,58]. In the case of hBN, Li et al. reported a tailored ball milling process (using benzyl benzoate as a milling agent) to induce gentle shear forces that result in high-quality hBN NSHs with a fair yield. This method can be applied to any layered material for producing NSHs [50].

Referring to chemical exfoliation methods, the viability of nano-manufacturing of graphene oxide (GO) NSHs was demonstrated via solution techniques, retaining the properties of bulk GO (thermal conductivity, mechanical properties, and optical transparency, among others) [59,60,61]. Amadei et al. [62] reported the synthesis of GO-NSCs using low- and high-frequency ultrasound solution processing techniques following a modified Hummers’ procedure [63], as illustrated in Figure 2. Fine-tuning of the obtained nanostructure dimensions, along with their surface chemistry modification, was achieved.

Recent work focused on acquiring hBN NSHs by chemically exfoliating bulk hBN powder. Lee et al. reported the exfoliation and functionalization of hBN by ball milling in an aqueous NaOH solution [64]. The hydroxide-assisted ball milling enhanced the yield and dispersibility of hBN. In other work [65], hBN particles were dispersed in NaOH solution and exposed to a pulsed ultrasound resulting in increased yield of hBN NSHs. The exfoliation and self-curling of hBN NSHs were also investigated by Li et al. [51], where they developed a low-temperature process to exfoliate hBN and produce NHSs, by heating a mixture of hBN powder, KOH, and NaOH in a Teflon-lined autoclave at 180 °C. In this work, the authors demonstrated the insertion of cations (Na^+^ or K^+^) and anions (OH^−^) into the interlayer spaces. When the coverages of the ions become high enough (≥10%) on the two surfaces of a BN monolayer, the self-curling energy exceeds the interlayer binding energy, favoring the peeling off from the BN bulk. Examples of the obtained nanostructures are shown in Figure 3.

More recently, a two-step method for the simultaneous formation and functionalization of hBN NSHs and NSCs was presented [66]. The first step consisted of a conventional chemical exfoliation under alkaline conditions [51], and the second step was the exposure of the obtained NSHs to low-frequency ultrasonic irradiation. The authors successfully obtained NSHs and NSCs, and then demonstrated the effective incorporation of O atoms to hBN nanostructures. In addition, the hBN NSHs undergo a phase transition from sp^2^ bonding to a mixture of sp^2^ and sp^3^ bonding under ultrasonic irradiation. This phase transition was attributed to the asymmetric vibrations and bending deformations of hBN NSHs upon scrolling to form NSCs [66].

### 2.2. Bottom-Up Synthesis

Materials in nature are designed in a bottom-up manner, employing the self-assembly of fundamental building blocks, evolving over thousands of years via natural selection for remarkable attainment. They are sustainably developed in a hierarchical way to be lightweight with agility and elasticity, holding biotic and abiotic resistance. To imitate the complexity and function of nature’s materials, there are many examples in different fields and disciplines. For example, synthetic biologists have been inspired to build biomolecules such as nucleotides, proteins, and peptides, as artificial nanostructures that are made of synthetic materials [67]. In the field of nanomaterials and nanotechnology, bottom-up syntheses are used to produce complex nanostructured materials from atoms and molecules, and they can often be tailored to control both the chemical and physical properties of the resulting nanostructures. These methods include gas-phase (physical vapor deposition (PVD) and chemical vapor deposition (CVD)) and liquid-phase reactions (sol–gel, Langmuir–Blodgett, thermal decomposition, and electrodeposition, among others). The liquid- and gas-phase syntheses have different time-scales; the slower liquid-phase processes can be used to obtain thermodynamically controlled products, while, for gas-phase synthesis, kinetic control is often the only option available [44].

Within the context of exploring and mastering the synthesis of advanced materials, different intents for the production of smart NSHs have been reported. Herein, the example of bottom-up fabrication of NSHs reported by Weiming et al. is presented [68]. The authors produced microscrolls by rolling up bilayers of polyvinyl alcohol (PVA)–poly(acrylic acid) (PAA)/metal (Ti, Au, Cr, Ag), which were prepared by evaporating the metal layer on top of a thermally crosslinked PVA–PAA layer (Figure 4a–d). The stimulus-responsive behavior of the microscrolls, caused by the volume change of the PVA–PAA hydrogel in response to environmental fluids, was the origin of the observed rolling and unrolling mechanism. The strain within these nanostructures was caused by the swelling of the PVA–PAA hydrogel in aqueous media [68]. Kim et al. explored a new synthetic design that controls the way in which nanomaterial building blocks adjoin to form the backbones that create tiny biomimetic sheets, using a bottom-up architectural design based on polymer templating [5,24,69]. The example illustrated in Figure 4e shows a thin film (e.g., gold or carbon) deposited via a gas-phase sputtering-based technique on a solid substrate. However, it is very difficult to exfoliate the micro- or nanosheets from the substrate because the process requires a good control over the interfaces between the different materials involved, and it is very hard to control or to fine-tune the shape due to a spontaneous rolling-up mechanism (Figure 4f), inhibiting stable water-dispersible free-standing NSHs. The intrinsic hydrophobic surface nature of metallic NSHs hinders their water stability, structural integrity, and biocompatibility. Surfactant-based exfoliation methods are often used but require a large volume of surfactants, which are usually challenging to rinse out. Without using surfactants, Kim et al. [69] synthesized readily exfoliable, biocompatible, and water-dispersible NSHs. For this purpose, the authors selected a synthetic biocompatible polymer (e.g., polyethyleneimine (PEI)) as a liquid exfoliation template and demonstrated a generalized, simple, and rapid supramolecular 2D self-assembly methodology at ambient conditions. These NSHs are in situ functionalized by adopting a water-soluble polymeric functionalization strategy, as shown in Figure 4f [5,24].

Of note, bottom-up approaches are more advantageous, as they allow producing nanostructures with fewer defects, more uniform chemical composition, and better scope for designing and ordering the different components, thus enabling a better control of the scalability of the synthesis processes.

### 2.3. Summary Points

Within the scope of this review, Figure 5 summarizes the two principal approaches considered for the synthesis of the NSHs. The top-down approach (Figure 5a) allows the production of free-standing NSHs, which can be subsequently functionalized via wet chemistry using surfactants, polymers, or biomolecules. The bottom-up approach instead (Figure 5b) allows both in situ functionalization and posterior functionalization of the NSHs.

## 3. Examples of NSHs Produced by Bottom-Up Methods

This section provides specific examples using a polymer-based templating approach. Different combinations of organic and inorganic materials are discussed. Spin-coating was the technique used for the polymer coatings, while the deposition of the inorganic thin films was performed using gas phase techniques, mainly sputtering and DC arc discharge [5,24,45,69].

### 3.1. Colloidal Smart NSHs for Biomedical Application

Creating smart NSHs would be a matter of combining the right materials. For example, NSHs made of or combined with a heat-sensitive polymer could burn surrounding tumors, functioning as a specific chemotherapy. When the appropriate (or specific) biomarkers are attached to the NSHs, the body sends them to the targeted tissues. In addition, the rolling process allows the NSHs to entrap drugs securely inside the frame and, thus, under external stimuli, release them to very specific regions of the body, which could minimize the amount of the drug necessary and reduce side-effects. Therefore, designing smart NSHs, which shift form in response to an external stimulus, can be used for a number of new applications [70].

Polymeric NSHs are promising and essential components due to their diverse organic functional groups. They have been extensively studied [71,72,73,74,75]. However, the emphasis of this review is on multicomponent inorganic NSHs, and hybrid NSHs (combination of inorganic–organic materials); thus, organic NSHs are not discussed in this work.

#### 3.1.1. Hybrid NSHs

Within this context, it is worthy to mention the work reported by Kim et al. [69], where the authors developed a new type of optically traceable gold-based NSHs, which respond to two different external stimuli: (i) acidity or basicity of their surrounding environment (pH); (ii) near-infrared (NIR) light, a wavelength of light that is harmless to human tissues. For the experiment, as depicted in Figure 6, a relatively simple polymer that responds to pH (polyethyleneimine (PEI)) was functionalized onto gold NSHs (previously deposited by physical vapor deposition (PVD) method). The resulting hybrid NSHs would regularly roll in basic, high-pH conditions, but remain planar in an acidic environment. This behavior offers the option for loading medicines inside of scrolled NSHs such that, when the sheets unroll, they release the drug. The pH-responsive NSHs, for instance, could be beneficial for targeting different parts of the human digestive tract, which changes pH between the acidic stomach and basic intestines. In addition, one can track the NSHs in vivo noninvasively, estimating pH [69].

#### 3.1.2. Janus NSHs

Another interesting type of NSHs, commonly known as “Janus NSHs”, consists of layered materials with high anisotropy [71]. Janus NSHs can be composed of layers of (i) inorganic materials (e.g., Au, MoS_2_, BN, and graphene), (ii) organic materials (e.g., polymers and surfactants), and (iii) a combination of both organic and inorganic materials, which are commonly referred as “hybrid Janus NSHs”. To develop hybrid Janus NSHs, the choice of the appropriate polymers is crucial due to their diverse organic functional groups. However, while Janus NSHs consisting of polymers have been extensively studied [5,51,52,53,54,55,72,73,74,75], only an extremely limited number of inorganic or hybrid Janus NSHs have been reported [47,48,49,76,77,78]. Hybrid Janus NSHs, with one hydrophilic face and one hydrophobic face, can be successfully synthesized via sequential surface modification of inorganic NSHs (e.g., gold) using different polymers, with subsequent exfoliation in aqueous solution. Kim et al. [79] demonstrated that 2D gold–polymer Janus NSHs can be fabricated using a simple layer-by-layer-based technique via a combination of physical vapor deposition (PVD) and polymer template liquid exfoliation (PTLE) techniques. The NSHs consisted of Au ultrathin films sequentially combined with three different biocompatible polymers (gum arabic, chitosan, and poly(ε-caprolactone)-b-polyethylene glycol (PEG-b-PCL)), resulting in nanocomposites that respond to pH changes and near-infrared (NIR) light, as shown in Figure 7 [79].

The heat-sensitive PEG-b-PCL is covalently anchored on the chitosan side of the NSH face (Figure 7b); thus, the NSHs are securely self-scrolled when hydrophobic anticancer molecules, e.g., doxorubicin (Dox), are loaded. When exposed to NIR irradiation, the formed scrolls undergo an unscrolling process to release the medicine with insignificant premature release [79]. Furthermore, the scrolled Janus NSHs show exceptional photothermal stability under NIR laser irradiation and a photo-hyperthermal effect capable of inducing death of cancer cells (e.g., HeLA), as well as two-photon bright fluorescence imaging, enabling potential cancer theranosis with minimal toxic side-effects [79].

### 3.2. NSHs for Molecular Sensing Enhancement

In this subsection, carbon-based NSHs decorated with Fe–Ag magneto-plasmonic nanoparticles (hereafter NSHs-NPs) are used as an example for low-concentration biomolecule detection. The biomolecule selected for this study is the ATP, a nucleoside triphosphate essentially used in cells as a coenzyme of intracellular energy transfer [80]. The technique used for the detection of Raman spectroscopy.

#### 3.2.1. Preparations of the NSHs-NPs

C-nanomaterials, such as nanotubes (NTs), nanowires (NWs), graphene oxides (GO), and carbon nanoparticles (NPs), have been widely developed as nanoscale sensors and biosensors due to the critical need for achieving better accuracy and sensitivity in point-of-care diagnostics [81]. According to theoretical studies, carbon NSCs can be used as a water channel model, enabling their use as ion channels across cellular membrane in biological systems or ion-controllable membrane filters [82]. Recently, graphene NSCs have shown potential in DNA sensing simulations due to their liquid gate structure, high conductivity, high current density, and switching capabilities [83].

Within bottom-up methods, only few studies have referred to the synthesis of carbon NSHs. For example, Turchanin et al. [84] self-assembled biphenyl molecules on a substrate from a solution, and subsequently crosslinked them by electron irradiation. The resulting NHSs were used for tunable conductivity purposes. A new approach to generate carbon-based NSHs and NSCs by combining vapor phase synthesis techniques with PTLE protocols was reported [24]. For this purpose, the first step was to spin-coat branched polyethyleneimine (bPEI) on a clean surface. Then, the carbon thin film was deposited on the resulting bPEI surface via the condensation of clusters of C-atoms, produced by the DC arc discharge technique. Subsequently the NPs were produced using an inert gas condensation method [24] and in situ landed on the surface of the carbon film. The NPs are composed of a Fe core and Ag shell, encapsulated with a Si outer shell (referred as Fe@Ag@Si). Figure 8a shows a scheme of the obtained layered structure. Fe@Ag@Si NPs supported on the top of the carbon film are represented by the blue small balls. Figure 8b,c show TEM and STEM images, respectively, of the obtained Fe@Ag@Si NPs embedded in carbon film. The details of synthesis protocols and characterization of these materials can be found in [24,85].

#### 3.2.2. Evaluation of NSHs-NPs for ATP Detection

The use of Raman spectroscopy in fast and sensitive detection is still limited because of the extremely weak signal. Surface-enhanced Raman scattering (SERS) techniques have emerged as sensitive detection techniques with high levels of molecular specificity, especially in the field of biosensing [86]. Figure 8a displays the experimental setup used for this evaluation. Briefly, the pH of ATP solutions was adjusted to 5.8 with phosphoric acid. The solution was then dispersed to the sample of carbon NSHs-NPs [24,85].

Figure 8d shows the Raman spectra of ATP at different concentrations ranging from 10^−7^ to 10^−10^ M. The prominent Raman peaks of ATP molecules at ~730 and ~1328 cm^−1^ are commonly attributed to the ring-breathing of adenine rings and C5–N7 stretching, respectively [80]. The enhancement factors (EFs) estimated from the peak at ~730 cm^−1^ are ~1.45 × 10^8^ and ~7.94 × 10^5^ for 10^−10^ M and 10^−7^ M, respectively [24]. The EF values can be attributed to the interaction of the incident electromagnetic field with surface plasmon resonance on the surface of the NPs (especially the Ag shell) and at the interparticle junctions of the plasmonic shell (Ag), resulting in plasmonic hotspots. These EF levels are encouraging and may be an advance toward single-molecule detection [87,88] or an ultrafine catalytic reaction [89].

## 4. Mechanisms behind the Scrolling and Unscrolling of NSHs

Herein, the aim is to discuss the process of the scrolling up of the NSHs independently of the method of their production. Generally speaking, NSHs can self-assemble via folding, rolling, bending, curling, or other shapes to form 3D objects. Strain-driven nanoarchitectures with a particular focus on metal–polymer bilayers and group IV semiconductors were reported by Li et al. [68] and Huang et al. [90], respectively. The mechanisms of scrolling of the NSHs to form NSCs can be divided into different categories depending on the surface functionalization and the structure of the NSHs.

### 4.1. Inorganic NSHs

Nanosheets were observed to curl up and form NSCs under sonication, e.g., the case of graphene and hBN [62,66]. While NSHs are exposed to ultrasonic irradiation, the ultrasonic energy imparted to the surrounding medium leads to the activation energy of the NSHs to form NSCs. In the specific case of hBN, Perim et al. [91] suggested that, when the NSHs are rolled up into NSCs, the layer curvature, torsion, and inversion increase and contribute to the strain energy; subsequently, the van der Waals interactions of the overlapping surfaces of the rolled layers contribute to the structural stability.

Other mechanisms of scrolling were induced by depositing magnetic nanoparticles on the surface of graphene oxide NSHs. The authors demonstrated that the rolling is initiated by the strong adsorption of maghemite nanoparticles at nitrogen defects induced in the graphene lattice and their mutual magnetic interaction [92]. Another study attributed the scrolling up of NSHs of graphene, hBN, MoS_2_, and TS_2_ to van der Waals interactions [93]. In this work, AgCN nanoparticles in situ nucleate and grow on the edges of the NSHs driven by the activated dangling bonds on the edges of the NSHs [44,93]. The resulting AgCN would change the electron density on the surface of the graphene, increasing the surface energy of the graphene. Thus, when the surface energy of the graphene accumulated to a high level with the increase of the forming AgCN particles (Figure 9e), the NSHs were triggered to curl up to reduce their surface energy. Therefore, the van der Waals interactions between the walls take over, resulting in the complete scrolling up of the NSHs to form NSCs (Figure 9f) [93].

### 4.2. Hybrid Inorganic–Organic NSHs and Janus Hybrid NHSs

In the case of NSHs made of combinations of organic and inorganic materials, independently of their category (composites, layered, or Janus), the mechanisms of scrolling and unscrolling depend basically on the nature of the polymers used to obtain the stimulus-responsive reactions. For example, combining polymers sensitive to the pH of the surrounding medium with inorganic materials, as indicated in Figure 10a, leads to NSHs that shrink (or expand) as the pH of the surrounding medium changes, inducing the rolling up or (or unrolling) of the NSHs. For example, natural polymers, such as polysaccharides, containing carboxyl groups or amino groups respond to the pH changes by changing their volume in the protonation/swollen state [94,95]. At low pH values, cationic polymers containing amino groups exhibit a stronger ionization and, therefore, higher swelling. On the contrary, the carboxyl containing anionic polymers show minimum ionization and, therefore, reduced hydration. When the pH is above the pKa of chitosan (~6.5), the amino groups start to become deprotonated, leading to polymer shrinking and “one-sided scrolling” of NSHs, while, at a pH below the pKa, the amino groups are protonated, which results in polymer expansion/swelling and, therefore, unrolling of formed NSCs.

Therefore, the combination of two polymers with opposite sensitivity to the pH of the surrounding medium, e.g., the case of the combination of chitosan (Chi) and gum Arabic (GA) [69,79,96], would result in responsive NSHs as described in Figure 10b. Chi and GA were selected as the smart polymers that expand/shrink in protonated states (lower pH values) and shrink/expand in non-protonated states (higher pH values), respectively [69,79,96].

## 5. Concluding Remarks

Although the phenomenon of exfoliation was firstly demonstrated more than a half century ago, it was not at the center of research interests until recently. This review gave insight into the last advances in the investigation related to PTLE-based manufacturing of hybrid NSHs (or hybrid 2D nanostructures), and their shape-shifting capability to form NSCs (or 1D nanostructures), with emphasis on their potential in bioapplications. Herein, the world “hybrid” encompassed (i) multicomponent inorganic NSHs, (ii) organic–inorganic composite NSHs, and (iii) multilayered organic–inorganic nanostructures.

However, despite the rapid pace of advance in the arena of NSHs, these nanostructures must be carefully evaluated for biocompatibility in order to be relevant for in vivo biomedical applications such as targeted drug delivery, imaging, and therapeutics. For example, their stability over time is still questioned, and their physiological interactions with living tissues are not as well understood as in the case of relatively well-studied 2D materials such as graphene or hBN. Moreover, the complexities of their structure (morphology), compositions, variable particle size and shape, impurities from manufacturing, and protein and immune interactions result in a huge mishmash of knowledge on the biocompatibility of these materials. In the case of in vitro applications, the use of real-time diagnostic platforms or biosensors without medical observation is still a big challenge.

On the other hand, this topic shows how collaboration among physics, chemistry, and biomedicine enables the translation of exciting new technologies into clinical biomedicine, to improve the lives of patients. However, the “know-how” to handle and process NSHs for clinical applications (e.g., drug delivery systems) is still in its infancy. Although many nanomaterial-based drugs have entered clinical practice, and more are being investigated in clinical trials for a wide variety of indications, nanomaterials also face challenges, such as the requirement for better characterization and quality control, possible toxicity issues, and the lack of explicit regulatory guidelines. Therefore, interdisciplinary academic explorations should push forward and come up with new synthetic techniques, improved analytics, and better understanding of the physiological interactions of nanomaterials and NSHs with living tissues.

Currently, within biosensing and similar emergence of the bio-diagnostics field, the interest in products such as DiY (“do it yourself”) goods, the Internet of things (IoT), or wearable/transplantable sensors coupled with big data or artificial intelligence technologies continues to grow; thus, these products are coming into play for more advanced and smarter biosensors. Hence, the use of NSH- and NSC-based biomedical systems is expected to be greatly synergized to address current medical challenges.

## Figures and Tables

**Figure 1 molecules-28-01020-f001:**
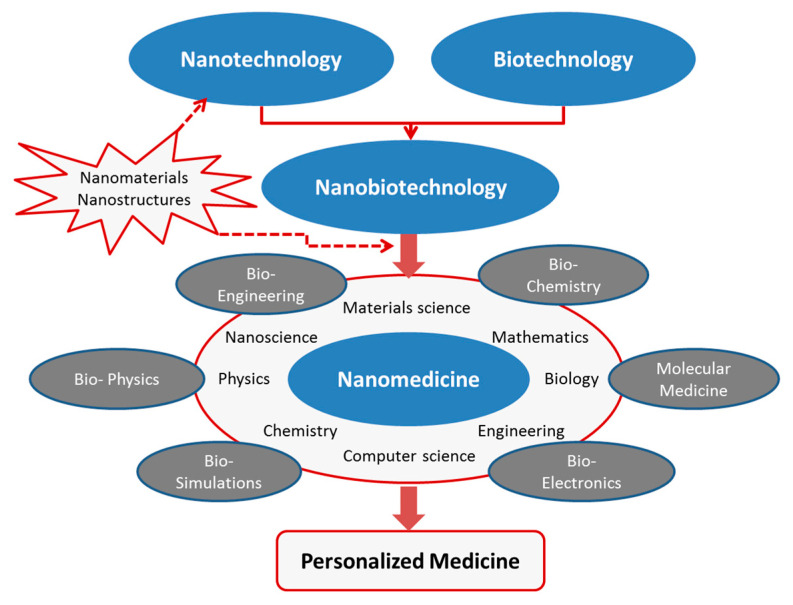
Contribution of nanomaterials and nanostructures to the multidisciplinary field of nanomedicine.

**Figure 2 molecules-28-01020-f002:**
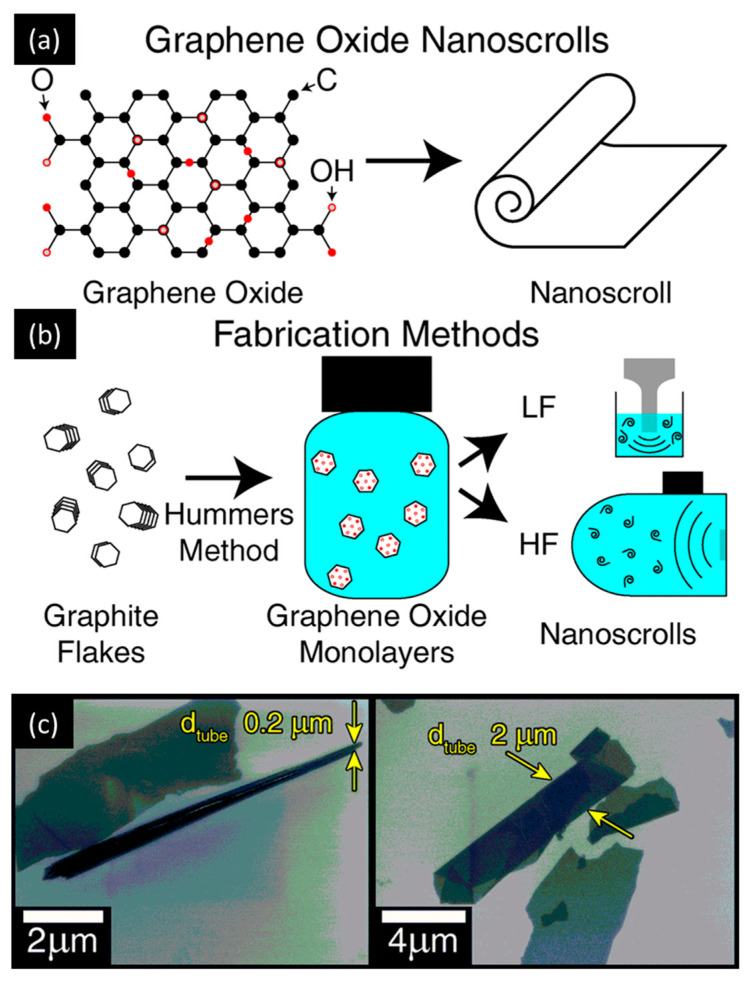
Structure and fabrication methods of GO and GO-NHSs. (**a**) Illustration of the GO-NSH chemistry showing the arrangement of the oxygen functional groups, and the cross-sectional morphology of GO-NSCs. (**b**) Summary of the GO-NSH fabrication process starting from graphite flakes and demonstrating the difference between the low-frequency (LF) and high-frequency (HF) processing. (**c**) Morphology of GO-NSHs: SEM displaying narrow and wide tube-like GO-NSCs that are characterized by their diameters (d_tube_). Adapted from Ref. [62].

**Figure 3 molecules-28-01020-f003:**
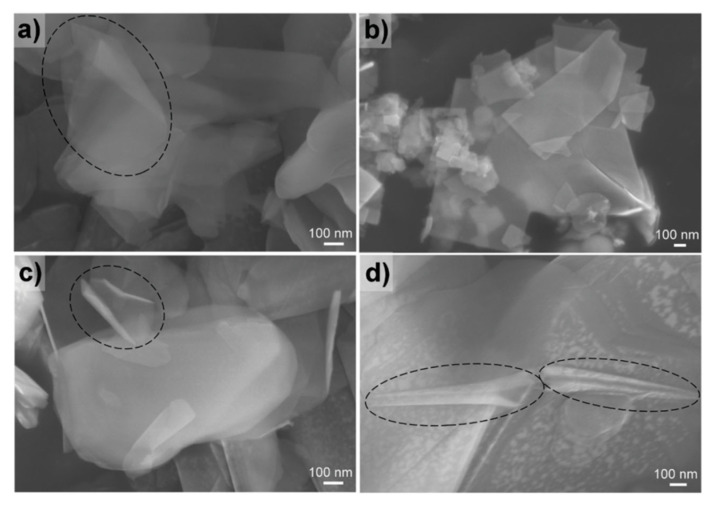
SEM images of hBN NSHs. (**a**) Agglomerated NSHs with several hundred nanometers in size (circled in black). (**b**) A very flat NSH. (**c**,**d**) Typical nanoscrolls (circled in black). Reprinted with permission from Ref. [51]. Copyright 2013, John Wiley and Sons.

**Figure 4 molecules-28-01020-f004:**
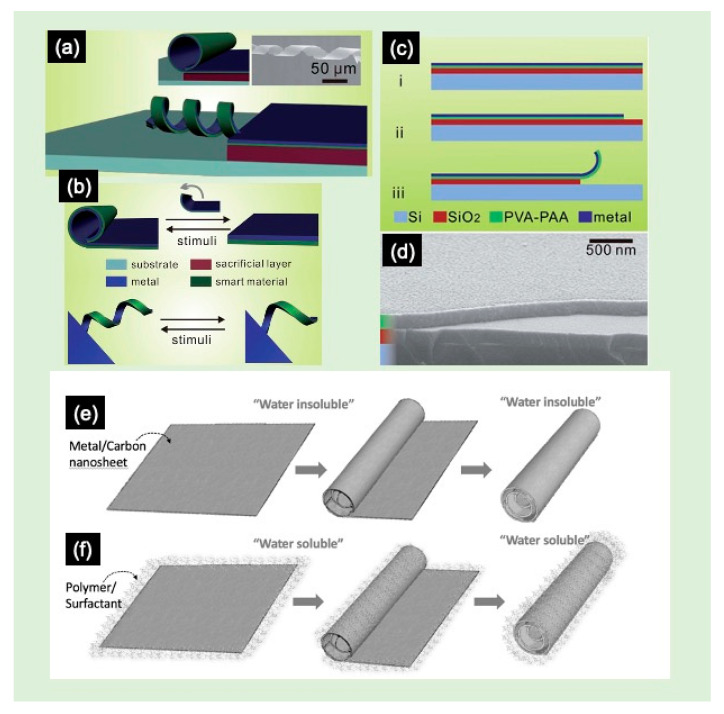
(**a**–**c**) Illustration of the basic process to fabricate bilayered micro-scrolls: (**i**) deposition of the polyvinyl alcohol–poly(acrylic acid) (PVA–PAA) and metal bilayer, (**ii**) lithographic patterning, and (**iii**) releasing process. (**d**) A cross-sectional SEM image of the PVA–PAA layer on the SiO_2_/Si substrate. The inset in (**a**) shows the SEM image of a titanium microspring fabricated using this technique. Adapted with permission from Ref. [68]. Copyright 2005, Royal Society of Chemistry. (**e**) Illustration of the fabrication steps of metal/carbon NSHs and NSCs, using polymer-free method. (**f**) Polymer-templated liquid exfoliation method. Adapted with permission from Ref. [5]. Copyright 2019, Elsevier.

**Figure 5 molecules-28-01020-f005:**
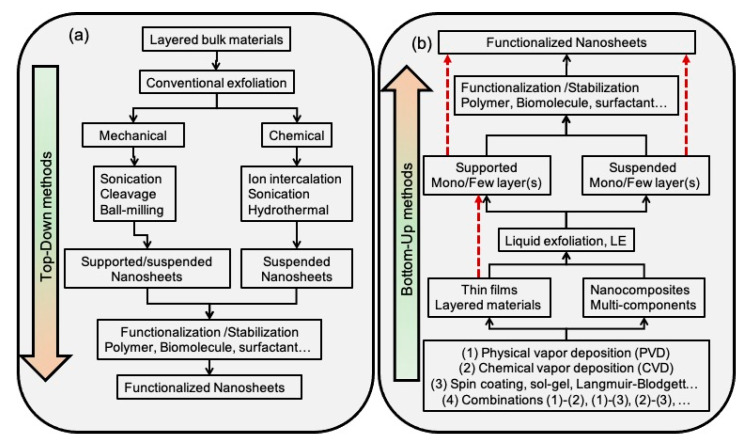
Scheme summarizing the bottom-up (**a**) and top-down (**b**) methods for the production of functionalized NSHs. With emphasis on inorganic, hybrid organic–inorganic, and Janus hybrid NSHs.

**Figure 6 molecules-28-01020-f006:**
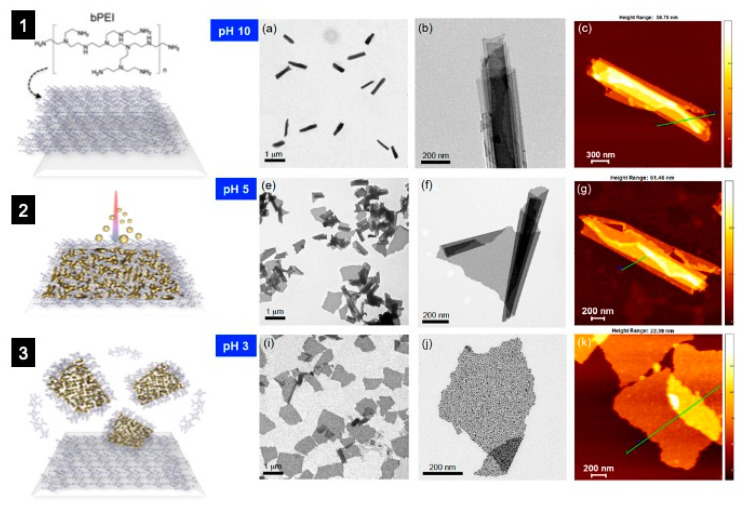
Schematic of the preparation and characterization of bPEI-Au NSHs. (**1**) The branched polyethyleneimine (bPEI) matrix was spin-coated onto a silicon wafer, (**2**) the Au film was deposited on bPEI/Si, and (**3**) the sample was dipped in methanol and exfoliated by ultrasonication. The pH-responsive behavior of the NSHs: (**a**–**c**) fully rolled NSHs at pH 10, (**e**–**g**) partially unrolled NSHs at pH 5, and (**i**–**k**) unrolled flat NSHs at pH 3. (**a**,**e**,**i**) TEM images showing rolled, partially unrolled, and fully unrolled NSHs, respectively. The scrolls are linear with no fragmentation. Higher-magnification TEM images showing controlled unrolling from (**b**) rolled to (**f**) partially unrolled and to (**j**) unrolled NSHs. (**c**,**g**,**k**) AFM topography images, consisting of the TEM data. Adapted with permission from Ref. [69]. Copyright 2014, American Chemical Society.

**Figure 7 molecules-28-01020-f007:**
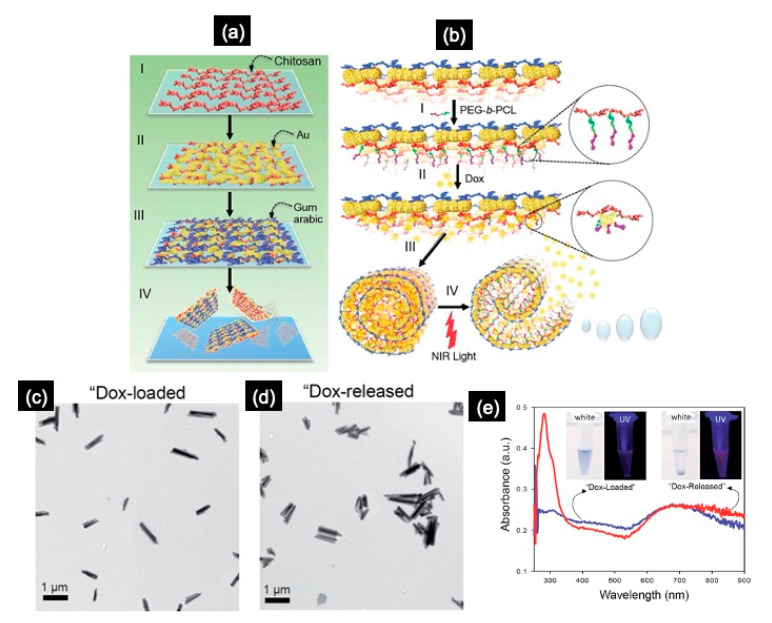
(**a**) Scheme illustrating the preparation of Janus composite Au-NSHs: (**I**) the chitosan matrix was spin-coated onto a glass wafer, (**II**) Au was sputtered onto the polymer-coated surface until a near-percolating Au thin film was obtained, (**III**) the Au surface was functionalized with gum arabic and (**IV**) dipped in acetate solution and exfoliated by ultrasonication, followed by purification and resuspension of the NSHs in deionized water. (**b**) The functionalization of poly(ε-caprolactone)-b-polyethylene glycol (PEG-b-PCL) block copolymers on the chitosan side of NSH (**I**), drug (Dox) loading (**II**), triggered scrolling (**III**), and NIR-induced unscrolling of a NSH resulting in drug release (**IV**) [79]. TEM images of nanoscrolls loaded with Dox before release (**c**) and after release by near-infrared (NIR) irradiation (**d**). UV/Vis absorption spectra of the corresponding Dox-loaded (blue) and released (red) nanosheet solutions (**e**). Adapted by permission from Ref. [79]. Copyright 2016, Royal Society of Chemistry.

**Figure 8 molecules-28-01020-f008:**
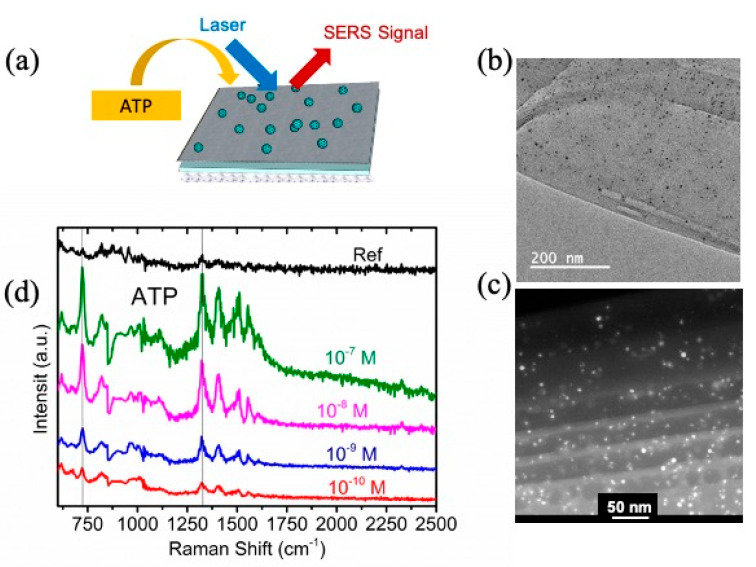
(**a**) Experimental setup for the evaluation of SERS-based biosensing performance using supported carbon NSHs-NPs. (**b**) TEM and (**c**) STEM images of carbon NSHs-NPs after exfoliation process, showing the distribution of Fe@Ag@Si NPs on the surface of carbon NSHs. (**d**) Concentration-dependent SERS of ATP deposited on supported NSHs-NPs. The “Ref” spectrum corresponds to 10^−2^ M ATP, which was dispersed directly on a Cu substrate. Reprinted with permission from Ref. [24]. Copyright 2016, American Chemical Society.

**Figure 9 molecules-28-01020-f009:**
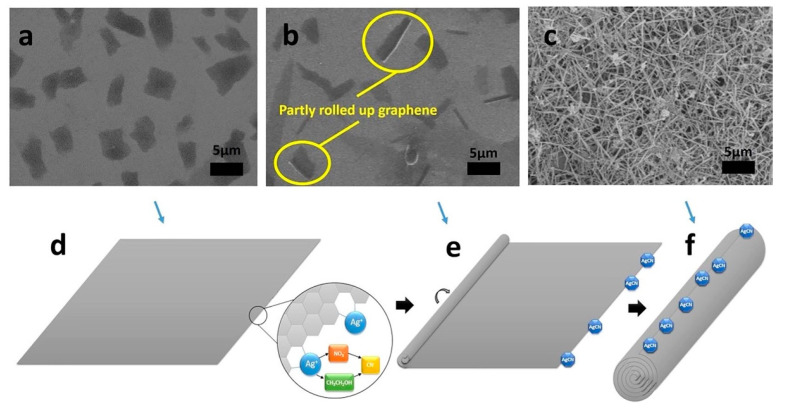
SEM images of the graphene NSHs (**a**), intermediate transition state (**b**) and the converted NSHs to NSCs (**c**). The speculation mechanism of the scrolling process of the graphene NSHs into the NSCs. A square graphene sheet and the position occupied with the created AgCN (**d**), the partially curled state of the graphene and the formed AgCN particles at the edge (**e**), and the final finished GNSs and formed AgCN particles on the surface of the GNSs (**f**). Reprinted from Ref. [93].

**Figure 10 molecules-28-01020-f010:**
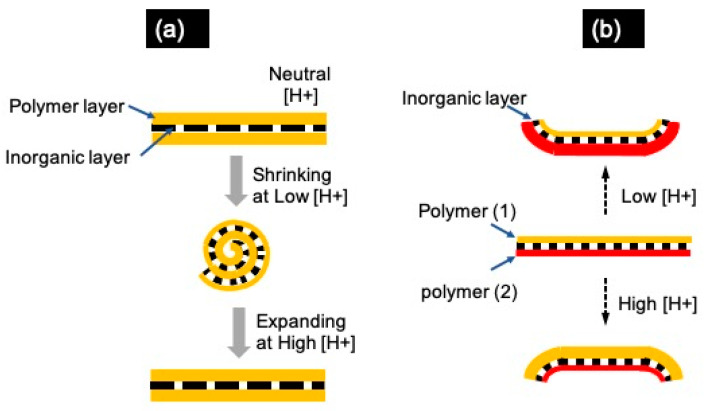
(**a**) Schematic scrolling/unscrolling mechanism of a hybrid NSHs, showing the reversible effects under various pH conditions. (**b**) A controlled scrolling/unscrolling mechanism of the Janus NSHs, comprising inorganic layers adsorbed onto two oppositely charged pH-sensitive polymer layers, presenting side-directed rolling in response to pH.

## Data Availability

Not applicable.

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
