# Peer review of "Stimulus-Responsive Ultrathin Films for Bioapplications: A Concise Review"

_molecules, 2023, doi:10.3390/molecules28031020_

Round 1
Reviewer 1 Report
The review summarizes interesting results and has a new outlook for nanosheets. The review can be accepted for publication after major revision. Some important issues should be clarified.
First, stimuli-responsive nanosheets should be compared to other types of stimuli-responsive ultrathin films such as grafted polymer brushes, Langmuir–Blodgett films, etc. The advantages and disadvantages of the bioapplication of the nanosheets should be taken into account.
An explanation on the stimuli-responsivity of the nanosheets is absent in the introduction. Why nanosheets are stimuly-responsive, and what leads to these mechanisms?
It is worth to present the most successful examples of synthesis and application nanosheets in the form of a table. It will help the readers to understand the essence of the topic.
Outlook in the future is absolutely absent in the conclusion, it is a very important part of the review because allows to understand the tendency of the development in this direction.
What does means the acronym PVA-PAA? (line 216)
Finally, I would like to recommend to cite the following review to compare responsive nanosheets with other similar systems:
https://doi.org/10.3390/polym14194245
https://doi.org/10.3390/pharmaceutics14112483
Reviewer 2 Report
This review focuses on nanosheets and their derivatives, nanoscrolls and Janus nanosheets. Synthesis of the 2D nanomaterials is classified broadly to top-down and bottom-up techniques, and their features and examples are systematically discussed. And possible biomedical applications using the smart 2D nanomaterials in response to various stimuli are introduced. As authors described, nanomaterials have been applied routinely for widespread fields, and an interest for 2D nanomaterial with unique structural properties is also increasing. Thus, this review is a valuable as a concise summary of nanosheet syntheses and applications. However, most of the research introductions in section 3 are author’s previous researches, and the content is very biased. In order to be accepted by a wide range of researchers involved in nanomedicine, the latest trends should be discussed from a multifaceted perspective, not only by the authors' research. The reviewer requests major revision in section 3 to recommend this review for acceptance. The other aspects that authors should be improved are following:
1. The description of Figure 1 in the text and the content of the figure are inconsistent. “Nanotechnology” and “biotechnology” are combined to develop into “nanobiotechnology” in this figure, but nothing is mentioned in the text. In addition, many of the research fields surrounding “nanomedicine” are included in “nanotechnology” and “biotechnology”, and the meaning of the hierarchy between the research fields in the gray oval and inside the red frame is also unclear (e.g. “physics” and “biophysics”, “chemistry” and “biochemistry”). Authors should clarify the relationship between the research fields in Figure 1, and explain their intentions carefully in the text.
2. Disadvantages of top-down synthesis are listed in lines 130-132, but there are no reason and related references. In discussing top-down synthesis, it is extremely important to understand the essential cause of the disadvantages, and the detailed explanations and appropriate literature should be provided. Similarly, the reasons and references should be also given for the advantages of bottom-up synthesis described in lines 247-250.
3. There is no mention of the nanosheet size throughout the review article. Lateral size, thickness, and aspect ratio are extremely important factors in discussing the structural properties of nanosheets. In addition, the size of nanocarriers in DDS is closely related to cell uptake and stealth properties, and their size-dependence for nanosheets have been also reported (e.g. Q.Mu et al., ACS Appl. Mater. Interfaces, 4, 2259 (2012)). The authors should describe the nanosheet size, its relationship with biomedical application, and size data in the referring researches.
Reviewer 3 Report
This review has major flaws and is poorly organized and written. Here are the comments:
1. There is a additional “Review” in the tile.
2. The references styles are full of inconsistency and format mistakes.
3. The language needs further improvements.
4. There are also many typos and mistakes in the main article. For example, line 383: error….
5. Where are the information about “Stimuli-responsive”?
6. The title says “ultrathin films”, but the article is full of nanosheets (NSHs), which one is your theme?
Reviewer 4 Report
Free-standing nanosheets made of organic, inorganic and hybrid materials are of great interest due to their large surface area with an abundance of available active sites, which promises numerous applications such as electronic and optical devices, membranes, drug delivery systems, etc. This review considers nanosheets of gold, amorphous-carbon, graphene, and boron nitride in terms of stimuli-responsive properties having potential for bio-sensing and controlled drug release systems. Methods for synthesizing such systems and devices based on them are considered in detail, including top-down and bottom-up approaches. It was shown that polymer-template-liquid-exfoliation methods are the most suitable as they lead to the fabrication of high-performance nanosheet composites and devices with the appropriate quality, solubility, and properties. As one of the variants of the nanostructures with stimuli-responsive properties, the authors consider the ability of nanosheets to roll into nanoscrolls (open tubular structures with adjustable interlayer spaces between their walls) which describes in details. Various examples of such systems are given.
I believe that the article is of interest for publication in the journal and can be published after the appropriate revision listed below.
1. Drug delivery and biosensing are considered important applications. Authors should more clearly write in the introduction section the criteria for use in such applications, for example, give the required dimensions of nanostructures, solubility in certain systems, etc.
2. It would be useful to present some of the described examples in the form of a table, including the type of system, material, synthesis method, type of response, reference.
3. One of the interesting object described in the review is folded nanosheets (nanoscrolls). However, the folding mechanism seems to be overlooked. Authors should provide possible folding mechanisms (see for example DOI: 10.1039/C0NR00648C, DOI: 10.1021/acs.chemmater.7b05324) that are important for subsequent applications.
4. References should be expanded. Since free-standing folded nanosheets are little known to readers, it is necessary to expand the list of materials. It is necessary to provide references to nanoscrolls of transition metal dichalcogenides (for example, DOI: 10.1002/smll.201601413), A2B6 monolayers (DOI: 10.1021/acs.chemmater.9b02927, DOI: 10.1021/acs.chemmater.7b05324), oxide systems (DOI: 10.1021/jp037200s).
5. It is not entirely clear from the text how Janus nanosheets are arranged: is one surface hydrophobic, the other hydrophilic? It is desirable to give a clearer description and give a scheme of Janus sheets.
6. The conclusion should be completed. It is necessary to consider further prospects for research in the field of nanosheets and applications of such systems.
Round 2
Reviewer 1 Report
The review was strongly improved after revision and can be accepted for publication in the present form.
Author Response
We thank this reviewer for his/her positive comments
Reviewer 2 Report
The authors improve the manuscript and answer reviewer’s questions.
Thus, I recommend publication in present form.
Author Response
We Thank this reviewer for his/her positive comments
Reviewer 3 Report
The corrected version can not fully address my questions. I insist my original decision.
Author Response
Following the Academic Editor’s comment, we have introduced the following modifications:
1- We added two new references:
- Reference [1], one of the best books in the field of nanostructures and nanomaterials, where thin films are defined/presented as 2D materials, and discussed from a such point of view. (Section 5: “2D-nanostructures: thin films”)
“Cao, G.; Wang, Y. Nanostructures and nanomaterials, 2nd ed, World Scientific publishing 2011”
- Reference [2] refers to metallic 2D-materials and discusses ultrathin metallic materials as 2D-materials:
“Ye, S.; Brown, A.P.; Stammers, A.C.; Thomson, N.H.; Wen, J.; Roach, L.; Bushby, R.J.; Coletta, P.L.; Critchley, Connell, S.D.; Markham, A.F.; Brydson, R.; Evans, S.D. Sub-Nanometer Thick Gold Nanosheets as Highly Efficient Catalysts. Adv. Sci. 2019, 6, 1900911”
- Reference [5] (ref 3 in the previous version) gives an insight into ultrathin 2D materials in the fields of condensed matter physics, materials science, and chemistry:
“Zhang, H. Ultrathin two-dimensional nanomaterials. ACS Nano 2015, 9(10), 9451-9469”.
2- These references are cited in the manuscript as follows:
“….Hence, the terms “2D materials”, “ultra-thin films”, and “nanosheets” are widely used as synonyms to describe materials that have one dimension in the nanoscale and are extended in the other two dimensions [1,2,5]….."
3- The references along the manuscript were updated (and highlighted in green) according to these modifications.